# Reducing the Permittivity of Polyimides for Better Use in Communication Devices

**DOI:** 10.3390/polym15051256

**Published:** 2023-03-01

**Authors:** Yuwei Chen, Yidong Liu, Yonggang Min

**Affiliations:** School of Electromechanical Engineering, Guangdong University of Technology, No. 100 Waihuanxi Road, Guangzhou HEMC, Guangzhou 510006, China

**Keywords:** fluorine, polyimide, dielectric properties, structural design

## Abstract

Recent studies have shown that introducing fluorinated groups into polyimide (PI) molecules can effectively reduce the dielectric constant (Dk) and dielectric loss (Df) of PIs. In this paper, 2,2′-bis[4-(4-aminophenoxy) phenyl]-1,1′,1′,1′,3,3′,3′-hexafluoropropane (HFBAPP), 2,2′-bis(trifluoromethyl)-4,4′-diaminobenzene (TFMB), diaminobenzene ether (ODA), 1,2,4,5-Benzenetetracarboxylic anhydride (PMDA), 3,3′,4,4′-diphenyltetracarboxylic anhydride (s-BPDA) and 3,3′,4,4′-diphenylketontetracarboxylic anhydride (BTDA) were selected for mixed polymerization to find the relationship between the structure of PIs and dielectric properties. Firstly, different structures of fluorinated PIs were determined, and were put into simulation calculation to learn how structure factors such as fluorine content, the position of fluorine atom and the molecular structure of diamine monomer affect the dielectric properties. Secondly, experiments were carried out to characterize the properties of PI films. The observed change trends of performance were found to be consistent with the simulation results, and the possible basis of the interpretation of other performance was made from the molecular structure. Finally, the formulas with the best comprehensive performance were obtained respectively. Among them, the best dielectric properties were 14.3%TFMB/85.7%ODA//PMDA with dielectric constant of 2.12 and dielectric loss of 0.00698.

## 1. Introduction

With the development of communication technology, the speed of communication waves in various fields are becoming faster, and the concept of 5G communication arises at the historic moment at which 3GPP designates 5G to support 450 MHz to 6 GHz and 24.25 to 52.6 GHz, which means that the communication wavelength has entered the era of millimeter wave [1]. Polyimide (PI) is favored by the electronics industry due to its excellent heat and corrosion resistance, electrical performance and processability, especially in the antenna group of communication receiving equipment [2,3]. When the insulating materials, such as PI, are under the effect of electric field, energy loss always occurs due to their conductivity and the polarization of the hysteresis. In addition, energy loss also occurs when electromagnetic waves pass through the insulation, which influences the electro-magnetic wave signal. Moreover, the energy loss increases with the increase of wave frequency. When it comes to the millimeter wave band, the energy loss will be very significantly increased. 

At present, the dielectric constant of commercial PI films on the market is above 3.0 (1 MHz), and the dielectric loss is above 0.01 (1 MHz) [4]. However, it can be predicted that, when commercial PI films are applied to 5G or even higher frequency communication equipment, the energy loss will increase exponentially, indicating that the commercial PI will have difficulty meeting the demand in the near future [5]. In recent years, many research teams around world have made non-negligible attempts to reduce the dielectric constant (Dk) and dielectric loss (Df) of polyimides [6]. Zeng et al. [7] studied the factors affecting the dielectric properties of polyimide by molecular simulation. Qiu et al. [8] prepared polyimide glass microspheres composite films and reduced the Dk to 2.26–2.48 by changing the spatial structure of the polymer. Many other groups also studied the effect of fluorine-containing groups on the dielectric properties of polyimide and prepared polyimides with Dk below 2.5 [9,10,11,12,13,14,15]. All the studies above were consistent with the simulation result gained by Zeng, and the research results show that the introduction of fluorine-containing groups is very beneficial to reducing the permittivity of polyimides. However, the modification can not be separated from the actual application demand, and other properties are ignored during the dielectric property modification [16].

According to the relationship between the structure and properties of polymers, it is found that the dielectric constant of PI is positively correlated with the polarizability, while the Df is related to the hygroscopicity. The higher the water absorption of the material, the greater the Df is [17,18,19,20]. The vitrification temperature is related to the pliability and interaction force of intermolecular chains [21,22], while the tensile strength is closely related to the chemical structure. By introducing the aromatic ring of the main chain, polarity of the side groups and hydrogen bonds could contribute synergetically to the increase of strength and chain rigidity [23,24]. Moreover, dense polar groups or substituents hinder the movement of chain segments, which cannot achieve high elasticity but make the material more brittle [25]. Furthermore, crosslinking can effectively enhance the connection between molecular chains and make them less prone to relative slip [26,27]. With the increase of the degree of crosslinking inside, it tends to be less prone to large deformation, which leads to the increase of its strength [28]. Furthermore, as the degree of molecular chain branching increases, the distance between molecules increases, and the intermolecular force decreases, so the tensile strength decreases, but the impact strength increases [29].

Many attempts have been made to simulate the polymer polarizability. Ankit Mishra’s group [30] introduced reactive force field (ReaxFF) into the model simulation. ReaxFF force field uses the concept of bond level to structure the bond and non-bond (including van der Waals and Coulomb forces) between all atoms in the system [31]. Rinaldi et al. [32,33] made a detailed derivation of the calculation formula and boundary conditions of the polarization from the theoretical modeling. Rivail [34] went on to improve some of the formulas, considering more detailed boundary conditions and electric field settings, while Bodo et al. [35] introduced the neglected differential diatomic overlap (NDDO) method into the previous theoretical derivation process and described the atomic charge calculation formula again.

Because both fluorine-containing groups and different structural symmetries can affect the polarity of Pi molecules, two fluorine-containing diamines with different structures and fluoride content were selected in this paper. In this study, 2,2′-bis[4-(4-aminophenoxy) phenyl]-1,1′,1′,1′,3,3′,3′-hexafluoropropane (HFBAPP) and 2,2′-bis(trifluoromethyl)-4,4′-diaminobenzene (TFMB) were selected as diaminobenzene monomer variables, diaminobenzene ether (ODA) as fixed diamine. 1,2,4,5-Benzenetetracarboxylic anhydride (PMDA), 3,3′,4,4′-diphenyltetracarboxylic anhydride (s-BPDA), 3,3′,4,4′-diphenylketontetracarboxylic anhydride (BTDA) were randomly copolymerized with diamines in ternary or quaternary composition. Firstly, the unit structure of PIs synthesized in this study was determined, and its model was imported into the software Material Studio 2019 to calculate the polarizability, and the law of the polarizability changing with the structure was obtained through analysis. Then the polymerization experiments were carried out, and the polyimide films were prepared by thermoacylation, and their mechanical properties, thermal properties, dielectric properties and moisture absorption properties were tested. Results showed that Dk first increased and then decreased as the fluorine content increased, and BPDA was better than BTDA for the reduction of Dk and Df. Finally, the best PI film formulas were found by comparison. In HFBAPP group, the comprehensive properties of 50%HFBAPP/50%ODA//PMDA was the best, with tensile strength of 72 MPa, elongation at break of 17.32%, glass transition temperature (Tg) of 351 °C, 5% weight loss temperature (T5%) of 540 °C, Dk of 2.39, Df of 0.00702 and hygroscopicity of 0.917%. In TFMB group, it was 14.3%TFMB/85.7%ODA//50%BPDA/50%PMDA, with tensile strength of 110 MPa, elongation at break of 20.23%, T5 of 568 °C, Dk of 2.12, Df of 0.00698 and hygroscopicity of 1.224.

The main contributions of this article are summarized in three aspects.

(1)This work uses a dual-line research path of simulation and actual experiment to study the related mechanisms and macro and micro performances of fluorine-containing groups affecting the dielectric properties of polyimide, and the theoretical and practical conclusions can be mutually verified.(2)This work focuses on more demanding communication usage scenarios, and specifically investigates the performance of the new PI films to meet the requirement of advanced communication technology.(3)The polymerization schemes with excellent performance are concluded, which also provide a variety of directions for further industrial application research of the prepared films.

## 2. Simulation Calculation

The molecular structures of PI that may be synthesized by ternary or quaternary polymerization in the experiments were determined, as shown in Figure 1 and Figure 2. Then the molecular structures above were imported into the simulation software Material Studio 2019 (MS, Accelrys, San Diego, CA, USA) to run the simulation.

Since the structures introduced in the simulation were the unit structures of each PI molecule, the volumes of the unit structure containing different amounts of monomers were not equal, so only the structures with the same number of monomers were compared simultaneously. For the above reasons, the single-chain structures of PIs in Figure 2 were chosen as the simulation models. One of them is shown in Figure 3, which is the model of PI polymerized by BPDA, ODA and HFBAPP.

The module ‘Forcite’ was adopted to optimize the structure, and it was also used to relax the simulation system by first using NVT with steptime of 600 ps under 298 K. This was to relax the molecular structure to minimize the energy. Figure 4 shows the changes of model energy and temperature during NVT process. Then NPT was used with steptime of 600 ps under 298 K and one standard atmosphere. This was to compress the molecules down to normal levels at one atmosphere for the convenience of subsequent calculation. Finally, the module “CASTEP” and “polarizability” were chosen to calculate the molecular polarizability, and the electric field is set by MS software.

It should be noted that, under ideal conditions, all monomers labeled as participating in polymerization were polymerized in the main chain, and only the molecular structure of the main chain covering all monomers was calculated, without considering the difference caused by the number of short chains in polymerization and the ratio of monomers. The cross-linking and crystallization during polymerization and imidization were preset (the cross-linking could not be controlled, and crystallization was avoided as far as possible through experimental heating process and cooling process design).

## 3. Experiment

### 3.1. Reagents

HFBAPP was procured from Meryer, Shanghai China, and ODA (99%), PMDA (99%), DMAc (99%), TFMB (98%), s-BPDA (98%), BTDA (98%) were obtained from Macklin, Shanghai China.

### 3.2. Ternary Polymerization of Different Diamine and PMDA

First of all, HFBAPP and ODA were randomly copolymerized with PMDA in five molar ratios (pure HFBAPP, 3:1, 1:1, 1:3, pure ODA) at suitable stirring speed. The molar ratio of diamines to dianhydride was 1:1.01, the solid content was 20% and the synthesis conditions were room temperature and 30% humidity. A series of PAA pre-polycolloids were obtained, and then the PI films were recorded as group HFOP after thermal imidization at 100 °C, 200 °C and 300 °C. Then TFMB and ODA were randomly copolymerized with PMDA at a molar ratio of 1:4–1:10 at an appropriate stirring speed. The experimental conditions and procedures were the same as above. The finished PI membranes obtained were denoted as group TFOP.

### 3.3. Effects of Different Types of Dianhydride

According to the experimental characterization results of the diamine group, we found that the molar ratios of the better comprehensive performance (see in the characterization results section for details) were 1:1 in the group HFOP and 1:6 in the group TFOP. Therefore, we chose these two proportions for the next experiment. The experimental process is shown in Figure 5.

Group HFBAPP: The molar ratio of HFBAPP to ODA was 1:1, and then the terpolymer of HFBAPP was carried out with BPDA and BTDA respectively. The molar ratio of diamine to dianhydride was 1:1.01, the solid content was 20%, and the synthesis conditions were room temperature and 30% humidity. A series of PAA pre-polycolloids were obtained, and then thermal imidization method was used. After high temperature treatment at 100 °C, 200 °C and 300 °C, the PI films obtained were denoted as group HFOB1. Then, quaternary polymerization was performed with PMDA, BPDA and BTDA (the molar ratio of dianhydride was PMDA:BPDA = 1:1, PMDA:BTDA = 1:1). The experimental process was the same as above, and the finished PI membrane obtained was denoted as group HFOB2.

Group TFMB: The molar ratio of TFMB to ODA was 1:6, and then polymerized with BPDA and BTDA, respectively. The molar ratio of diamine to dianhydride was 1:1.01, the solid content was 20%, and the synthesis conditions were room temperature and 30% humidity. A series of PAA pre-polycolloids were obtained, and then the PI films were de-noted as group TFOB1 after thermal imidization at 100 °C, 200 °C and 300 °C. Then, quaternal polymerization was performed with PMDA, BPDA and BTDA (the ratio of dianhydride was PMDA:BPDA = 1:1, PMDA:BTDA = 1:1). The experimental procedure was the same as above, and the finished PI membrane obtained was denoted as group TFOB2.

### 3.4. Measurements

First of all, in all measurements, three samples of the same film were used for each measurement, and each sample was measured five times. The data of the sample with the largest difference were removed and the average of the remaining four times was taken.

The Fourier transform infrared (FTIR) spectra of PI films were obtained using a Nicolet 6700 Fourier Transform Infrared Spectrometer. Thermogravimetric analysis (TGA) of the polymer nano composites were carried out with an TGA 4000 (Holland) at a heating rate of 10 °C/min from 30 °C to 750 °C under N2 atmosphere. Differential scanning calorimetry (DSC) of the composites was tested by a DSC 8000 (Britain) at a heating rate of 20 °C/min from 30 °C to 450 °C under N2 atmosphere. Tensile strength and elongation at break were obtained by Universal Testing Machine (America).

The thickness was obtained by vernier calipers (uncertainty of ±0.1 μm). Five points were evenly taken on the same sample to measure the thickness, and the average value was taken as the final thickness used. All thickness measurements were kept in two decimal places and in microns.

The dielectric constants and loss were determined by WY2858 Dielectric Spectrometer (Wuyi Electronic, Shanghai, China) in the range of 1 Hz to 1 MHz at 29% relative humidity. All of the films were cut into squares with the side length of 50 mm. The capacitance (uncertainty of ±0.1 pF) and dielectric loss (uncertainty of ±5% ± 0.0001) were obtained directly during measurement, and then the dielectric constant was calculated by the Equation (1). (The results of the calculation were reserved for two decimal places.)
(1)$$\varepsilon_{r}=\frac{C\cdot d}{\varepsilon_{0}\cdot S}$$
in which *C* is the value of capacitance, d is the thickness of films, *ε*_0_ is vacuum dielectric coefficient, *ε*_r_ is the relative dielectric constant and *S* is the measured area of the film.

The dielectric loss in this paper is the tangent of the dielectric loss Angle (tan*δ*) measured by the bridge method. At the time of measurement, the samples were in a parallel circuit, and Df could be calculated by the Equation (2).
(2)$$\mathrm{Df}=\frac{1}{\omega \cdot Cp\cdot Rp}$$
in which *ω* is the angular frequency (rad/s), *C*_p_ is the value of capacitance (pF), and *R*_p_ is the equivalent resistance of a parallel circuit (Ω).

The hygroscopicity of PI films was measured according to GB/T1033-1998. The details were as follows.

The PI films were dried in a 60-degree oven for 24 h. The mass m_1_ was then weighed. Then it was soaked in deionized water for 24 h and weighed immediately after wiping the water on the surface of the film to obtain the mass m_2_. Each sample was measured three times and averaged. The hygroscopicity can be obtained by the following formula.
(m_2_ − m_1_)/m_1_ × 100%(3)

## 4. Result

### 4.1. Results of Simulation

The final polarization results are shown in Table 1. Polarization result 1, Polarization result 2 and Polarization result 3 are three forms of polarization and were given by the result document. The three columns of data in the table represent, from left to right, the polarizability results obtained when the molecule applies an electric field to the cubic angstroms, cubic centimeters and atomic scales. The a.u. is a unit commonly used to add electric field in quantitative calculation, which can be converted to V/Angstrom. The conversion formula is as follows:1 a.u. = 51.423 V/Angstrom(4)

It can be seen from Table 1 that the introduction of fluorinated groups greatly reduces the polarization of molecular structure, and it confirms that the introduction of fluorinated groups is an effective method to reduce the DK and Df of PI. However, the data of line 11/12/13 show that if ODA is completely replaced with HFBAPP or TFMB, the polarization will increase. HFBAPP has a larger polarization than TFMB due to its larger main chain structure than ODA and TFMB, indicating that when fluoride content is certain, the polarization will decrease. For the ternary polymerization containing HFBAPP or TFMB, the polarization of PMDA is lower than that of BPDA and BTDA, and the Dk and Df of HFBAPP/ODA//PMDA and TFMB/ODA//PMDA will be lower. For the quaternion polymerization containing HFBAPP and TFMB, introducing BPDA and BTDA has little difference in the change of polarization, however, the polarizability of BPDA is relatively small, and the polarizability of both is lower than that of ternary polymerization.

### 4.2. Structural Elucidation: FTIR

According to the infrared spectrogram shown in Figure 6, four absorption peaks are observed at 716 cm^−1^, 1370 cm^−1^, 1716 cm^−1^ and 1780 cm^−1^, corresponding to bending vibration peak of C=O, stretching vibration peak of C-N and stretching vibration peaks of C=O respectively, which proved the completion of imidization. In addition, absorption peaks of C-F bond can be observed at 1173 cm^−1^, 1213 cm^−1^ and 1247 cm^−1^, indicating that -CF_3_ structure has been successfully introduced into PI molecule structure. According to the data above, it is found that fluorine-containing groups can be perfectly introduced into the main chain structure of polyimide without affecting the normal formation of the imide group.

### 4.3. Thermal Properties

The thermogravimetric curves are shown in Figure 7. Data in Table 2 showed that, after HFBAPP was added, the glass transition temperature increased, while T5% decreased. For all this, however, Tg were still above 320 °C and T5% were still above 490 °C, which means excellent thermal properties. Furthermore, the Tg of PI in HFBAPP:ODA = 1:1 was more than 350 °C, and the T5% was 540 °C, showing the best thermal performance in HFOP group. Table 2 also showed that the incorporation of BPDA could retain the excellent thermal performance of HFOP group to a greater extent. Although Tg decreased, it maintained at about 300 °C, and T5% was 540 °C at the same time, which met the needs in some industrial situations.

Figure 8 shows the TGA curves of PIs in group TFOP and TFOB. Table 3 shows that the thermal properties of PI films in TFOP and TFOB groups were excellent, and the T5% of PI films kept above 560 °C regardless of the changes of fluoride content and monomer components. However, PIs in this group have the problem that the glass state transition was not obvious. In the DSC test, even when the heating rate was raised to 20 K/min, no obvious glass transition interval could be observed (as shown in Figure 8c). According to the structure of TFMB and dianhydride, we preliminarily judged that it was probably because crosslinking produced in the polymerization, and high degree of crosslinking, to a certain extent, hampered the movement of molecular chain under high temperature, characterized as high thermo-gravimetric temperature, while hindering the PI films on the macro shift from glass state to the high elastic state, which made the change interval not obvious, or interval temperature so high as no obvious Tg point was observed in the conventional Tg interval.

From the data in Table 3, it can be seen that in the TFMB group, the T5% of PI film of TFMB:ODA = 1:6 is 568 °C, and the T5% of TFMB:ODA = 1:6/BPDA:PMDA = 1:1 is 587 °C. These are both excellent thermal performance and make the new PI suitable for many high temperature applications. In conclusion, from the perspective of thermal performance, adding an appropriate amount (25–50%) of HFBAPP appropriately increased the Tg of PI film, and adding TFMB increased the T5%. However, the drawback was that the Tg point was not obvious, so the two could be selected and proportioned according to the direction of demand.

It is confirmed that the introduction of fluorine-containing groups can indeed have a positive effect on the thermal stability of polyimide. However, according to polymer physics, the thermal stability of polymer is related to bond energy and crosslinking degree. The C-F bond has a relatively high bond energy, and the fluorinated diamines selected in this experiment all have three C-F bonds, which undoubtedly has a significant impact on the thermal stability. However, from the data in Table 2 and Table 3, they do not show an obvious linear relationship. It is speculated that this may be caused by the failure to control the molecular weight, main chain length and mass ratio of short chain, so that the macro thermal stability of each component is affected by many structural factors, and the influence mechanism is complex, and finally the difference in data is presented.

### 4.4. Mechanical Properties

It can be seen from Table 4 that the introduction of HFBAPP could indeed strengthen the mechanical properties of PI films at certain proportion points. Especially for 25%HFBAPP/75%ODA//PMDA and 50%HFBAPP/50%ODA//PMDA, the tensile strength of PI films was increased to 73 MPa and 72 MPa respectively, and the elongation at break were increased to 18.22% and 17.32%. The flexibility of PI film was greatly improved. According to polymer physics, the tensile strength of the polymer materials depends on the chemical bond force of the main chain and the intermolecular force. The introduction of HFBAPP made the molar ratio of aromatic heterocyclic in the unit structure of the modified PI material increase, which was reflected in the macroscopic improvement of the tensile strength and elongation at break of the modified PI films. However, when the proportion of HFBAPP increased, the effect of reduced polarizability on mechanical properties began to exceed that of aromatic hetero ring, and both tensile strength and elongation at break began to decrease.

In group TFOB, the PI films of BPDA ternary polymerization have become wrinkled and fragile after imidization at 350 °C, indicating that the molecular bonds inside have begun to break, which means that the thermal properties of the PI films of this formula cannot meet the requirements, so no performance test is conducted on it.

Due to the strong molecular structure rigidity of TFMB, it can be predicted that when the proportion of TFMB in the diamine system increases, the tensile strength will increase, and will begin to decline to a certain critical point. At this time, the PI film will be very brittle, and the elongation at break will decrease accordingly. Due to the pre-experiment, we found that when the ratio of TFMB and ODA was greater than 1:4, the PI films were relatively brittle. When the ratio was 1:4, there was no fracture of PI films after multiple folds. We judged that the mechanical properties of PI films within this ratio could be incorporated into further experimental plans.

As can be seen from Table 5, the tensile strength showed a trend of first increasing and then decreasing. As the tensile strength is closely related to molecular chemistry, there are many influencing factors. We speculated its rise because of the side chain—CF_3_ to strong polar bond, which drastically reduced the polarity of the molecular structure of the material. At the same time, although dense TFMB substituents made PI films brittle, the influence weakened as the TFMB content was reduced. When the ratio of TFMB:ODA was in the range of 1:4 to 1:6, the positive effect on tensile strength covered the negative influence of the latter. Then, as the content of TFMB in the diamine system continued to decrease, the proportion of side chain decreased, and the positive influence of tensile strength weakened, so the tensile strength began to decline. In group TFOP, the best ratio of tensile strength was TFMB:ODA = 1.6, the tensile strength was 110 MPa, and the optimal ratio of elongation at break was TFMB:ODA = 1:4, and the elongation at break reached 23.15%.

The data in Table 5 above also showed that, when PMDA was changed into BPDA or BTDA, the mechanical properties of PI material declined significantly. We speculated that the reason was the same as HFBAPP group, and since the molar mass of TFMB itself was much smaller than that of HFBAPP, when the average molar mass of PI increased, TFMB structure had a weaker effect on mechanical properties than HFBAPP. Similar to HFBAPP group, further change of the ratio of dianhydride did not have much effect on the mechanical properties of the material, and the tensile strength and elongation at break had no significant change and obvious trend. In the TFOB group, the best ratio of mechanical properties was 14.3%TFMB/85.7%ODA//50%BPDA/50%PMDA, at which the tensile strength was 56 MPa and the elongation at break was 9.23%.

### 4.5. Dielectric Properties and Hygroscopic

In order to reduce the thickness measurement error as much as possible, the thickness of the film was controlled at 25–30 μm. Then, the thickness of each sample was measured according to the measurement steps mentioned above, and the final dielectric constant value was calculated by substituting formula 1. It should be noted that the average thickness of each sample is not the same. Although the thickness affects the capacitance value, it is divided by the thickness at the same time during calculation. Therefore, for the same sample or samples obtained by the same experimental scheme, the final calculated Dk does not have much relationship with the thickness. Therefore, only the thickness is strictly measured, but the film thickness is not studied as an influencing factor.

As could be seen from Table 6, both Dk and Df decreased as the proportion of HFBAPP in the diamine system increased, which was consistent with the relationship of fluorine content in PI unit structure and dielectric constant and dielectric loss in the simulation calculation, and the hygroscopicity decreased first and then increased to within 1.5%. Water absorption of materials is related to hydrophilicity, hydrophobicity, porosity and pore size. The PI films prepared in this experiment did not have any pore-forming treatment, and the incorporation of air was also avoided in the preparation, and they were all hydrophobic films. The addition of fluorine atom enhanced the electronegativity of PI molecular structure, weakened the polarity, and further reduced the water absorption rate of PI films. However, there was no obvious relationship between water absorption rate and fluoride content. We conjectured that, when fluoride content increased to a certain extent, the influence on PI unit molecular structure was greater than the gain brought by electronegativity, leading to a certain increase in water absorption rate. According to the three data, when the ratio of HFBAPP:ODA was 1:1, the Dk was 2.39, Df was 0.00702, and hygroscopicity was 0.917%.

The data in Table 6 also showed that the dielectric properties and hygroscopicity of PI films obtained by ternary tetrad polymerization involving BPDA and BTDA were not different from each other. The performance of BPDA group was slightly better than that of BTDA group, but both were slightly worse than HFBAPP:ODA = 1:1 in Table 6, which was consistent with the polarization change result obtained by simulation calculation. The optimal ratio of dielectric properties was 50%HFBAPP/50%ODA//BPDA, with Dk of 2.38 and Df of 0.00734. The optimal ratio of hygroscopicity was 50%HFBAPP/50%ODA//50%BPDA/50%PMDA, which was only 1%.

Due to the pre-experiment, we found that the ratio of diamine in the HFOP group was not suitable for the TFOP group. When TFMB:ODA was 1:3, the synthesized PI films were very brittle. After the high temperature of 350 °C, the surface of PI films had obvious folds, which meant that the molecular structure broke at the beginning. Therefore, the TFOP group was tested from 1:4. As can be seen from Table 7, the dielectric constant and dielectric loss of TFOP group decreased with the increase of TFMB in the diamine system, reaching the lowest of 2.08 (TFMB:ODA = 1:4) and 0.00682 (TFMB:ODA = 1:6) respectively. It could be conjectured that the dielectric performance was ideal. Meanwhile, the hygroscopicity first decreased and then increased, and the lowest hygroscopicity of PI film in TFOP group was 1.224% (TFMB:ODA = 1:6). This was because when the diamine mole ratio started from 1:4, the fluoride content in PI perfectly avoided the inverse influence interval, which made the polarization decrease with the increase of fluoride content. Meanwhile, because the CF3 bond of TFMB was directly connected to the benzene ring, the molar mass of TFMB was smaller, and the fluoride content of TFMB was higher than that of HFBAPP and TFMB with the same molar amount. Moreover, TFMB had an asymmetric structure, which reduced polarity to a higher degree than HFBAPP, so it had a better improvement in dielectric performance.

In Table 7, when the ratio was 14.3%TFMB/85.7%ODA//PMDA, PI films had the best dielectric performance. At 1 MHz, the Dk was 2.12, the Df was 0.00732, and the moisture absorption rate was 1.224%. When the ratio of dianhydride was replaced, we found that the dielectric properties of BTDA and TFMB were better than that of BPDA, which showed a little difference with the simulation calculation result. This was because in the simulation, the effect of crosslinking on the polarization did not be considered, and crosslinking may occur in polymerization experiments containing TFMB. So we inferred that when TFMB polymerized with BTDA, crosslinking occurred, contributing to the result of dielectric properties of PI films prepared by polymerization with BPDA better than those prepared by polymerization with BTDA, and in the TFOB group, the best ratio of dielectric performance was 14.3%TFMB/85.7%ODA//50%BTDA/50%PMDA. At this time, the Dk of PI films was 2.25, and the Df was 0.00762, meanwhile moisture absorption rate was 1.128%.

In conclusion, in terms of dielectric properties, TFMB can reduce the dielectric constant of PI film more than HFBAPP, while HFBAPP is more suitable for introducing modified molecular structure than TFMB for hygroscopicity and dielectric loss.

## 5. Conclusions

In this study, three kinds of diamines (including two kinds of fluorinated diamines with different structures) and three kinds of dianhydride were selected to characterize the performance of PI films polymerized by them through simulation calculation and experiment. Since only the polarization of the structure is considered in the simulation calculation, and the factor is relatively single, the simulated change trend of the performance is slightly different from the experimental results, but generally consistent, and the difference can also be inferred according to the structure. The results show that the formula with the best comprehensive performance in HFBAPP group is 50%HFBAPP/50%ODA//PMDA. At this time, the tensile strength of PI film is 72 MPa, elongation at break is 17.32%, Tg is 351 °C, T5% is 540 °C, Dk is 2.39, Df is 0.00702. Hygroscopicity is 0.917%. The formulas with better comprehensive performance in TFMB group are 14.3%TFMB/85.7%ODA//PMDA and 14.3%TFMB/85.7%ODA//50%BPDA/50%PMDA. The tensile strength of PI film is 110 MPa, elongation at break is 20.23%, T5% is 568 °C, Dk is 2.12, Df is 0.00698, and hygroscopicity is 1.224. The latter PI film has tensile strength of 56 MPa, elongation at break of 9.234%, T5% of 587°C, Dk of 2.29, Df of 0.00786, and hygroscopicity of 1.274. In actual production, formulations with better performance can be selected for further experiments according to the target performance.

## Figures and Tables

**Figure 1 polymers-15-01256-f001:**
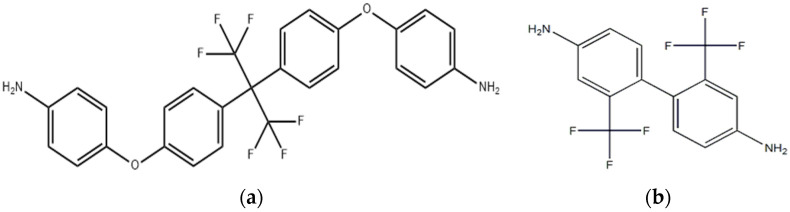
(**a**) Molecular structure of HFBAPP; (**b**) Molecular structure of TFMB.

**Figure 2 polymers-15-01256-f002:**
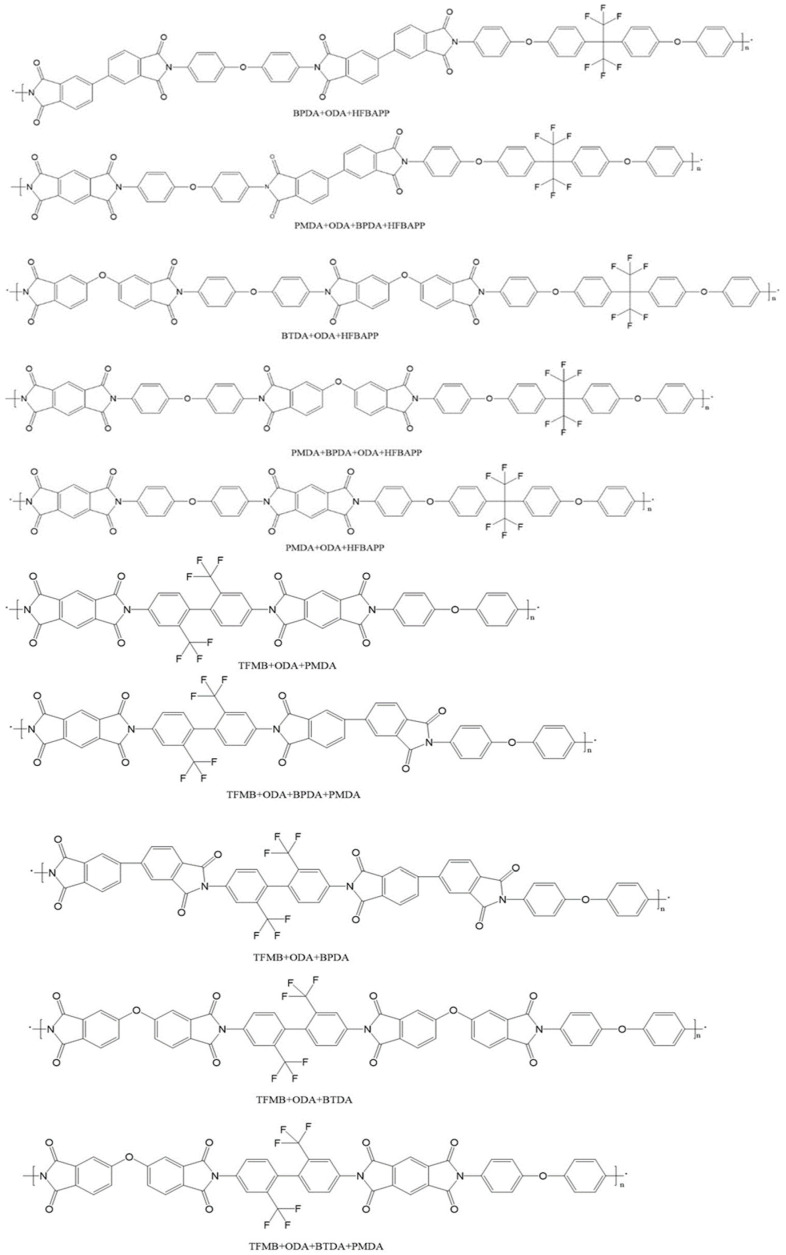
Molecular structure diagram of PIs of Group HFOB, HFOP, TFOB and TFOP.

**Figure 3 polymers-15-01256-f003:**
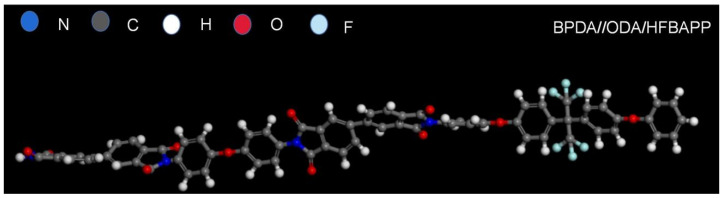
An example of models used for simulation: BPDA//ODA/HFBAPP.

**Figure 4 polymers-15-01256-f004:**
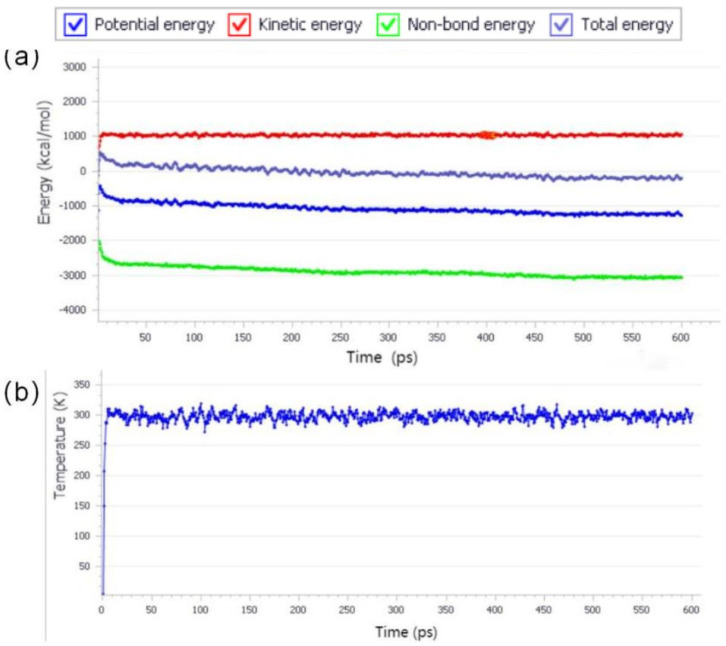
(**a**) Dynamic energy curves of model in NVT ensemble; (**b**) temperature curves of model in NVT ensemble.

**Figure 5 polymers-15-01256-f005:**
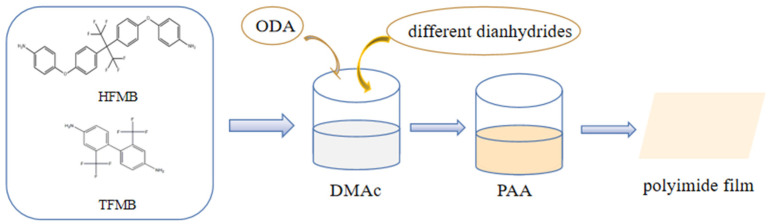
The synthesis process of fluorinated polyimides.

**Figure 6 polymers-15-01256-f006:**
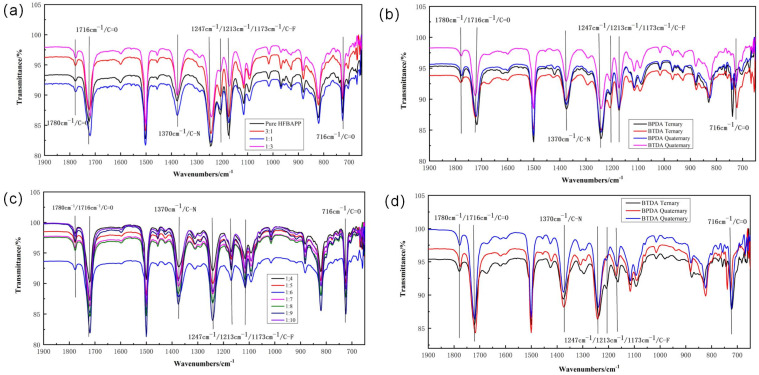
FTIR spectra of fluorinated PI films (**a**) FTIR spectra of group HFOP; (**b**) FTIR spectra of group HFOB; (**c**) FTIR spectra of group TFOP; (**d**) FTIR spectra of group TFOB.

**Figure 7 polymers-15-01256-f007:**
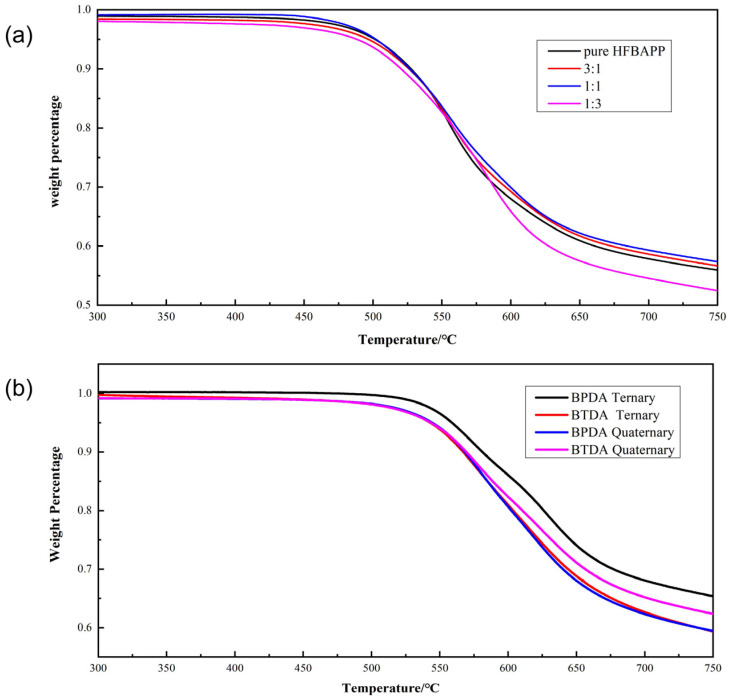
Thermal properties of films in Group HFOP and HFOB. (**a**) TGA curves of films in group HFOP (**b**) TGA curves of group HFOB.

**Figure 8 polymers-15-01256-f008:**
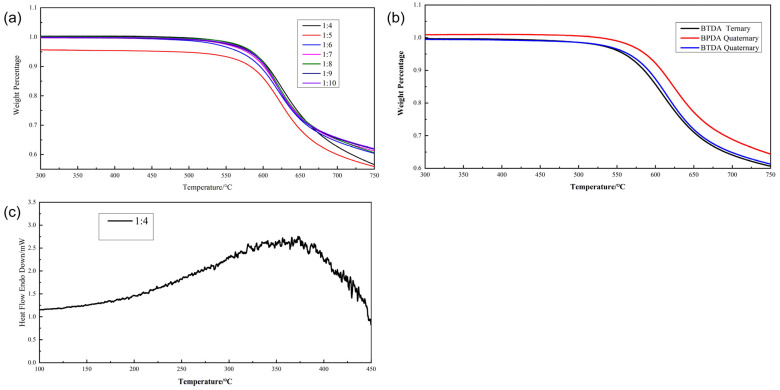
Thermal performance diagram of films in group TFOP and group TFOB. (**a**) TGA curves of films in group TFOP (**b**) TGA curves of films in group TFOB (**c**) DSC curve of films in PI film when TFMB:ODA = 1:6.

**Table 1 polymers-15-01256-t001:** Simulation results of the polarizability of PI films composed of each monomer group.

	Monomers	Polarization Result 1/Angstrom^3^	Polarization Result 2/cm^3^	Polarization Result 3/a.u.
1	ODA/HFBAPP//PMDA	105.30900	1053.09001	710.66097
2	ODA/HFBAPP//BPDA	124.33796	1243.37963	839.07488
3	ODA/HFBAPP//BTDA	125.96996	1259.69965	850.08819
4	ODA/HFBAPP//BPDA/PMDA	114.35725	1143.57253	771.72166
5	ODA/HFBAPP//BTDA/PMDA	115.77274	1157.72736	781.27381
6	ODA/TFMB//PMDA	83.82904	838.29040	565.70689
7	ODA/TFMB//BPDA	102.40759	1024.07595	691.08130
8	ODA/TFMB//BTDA	103.66158	1036.61576	699.54359
9	ODA/TFMB//BPDA/PMDA	93.11482	931.14817	628.37047
10	ODA/TFMB//BTDA/PMDA	93.33594	933.35938	629.86267
11	ODA//PMDA	41.06345	410.63453	277.11015
12	HFBAPP//PMDA	65.68756	656.87561	443.28201
13	TFMB//PMDA	43.29990	432.99903	292.20248

**Table 2 polymers-15-01256-t002:** Thermal properties of PI films in group HFOP and group HFOB.

Monomers	T_g_/°C	T_5%_/°C
ODA//PMDA	340	550
25%HFBAPP/75%ODA//PMDA	346	490
50%HFBAPP/50%ODA//PMDA	351	540
75%HFBAPP/25%ODA//PMDA	321	500
HFBAPP//PMDA	320	510
50%HFBAPP/50%ODA//BPDA	278	560
50%HFBAPP/50%ODA//BTDA	275	543
50%HFBAPP/50%ODA//50%BPDA/50%PMDA	297	540

**Table 3 polymers-15-01256-t003:** Thermal properties of PI films in group TFOP and group TFOB.

	Monomers	T_1%_/°C	T_5%_/°C
TFMB:ODA	1:4	531	582
	1:5	502	581
	1:6	488	568
	1:7	507	575
	1:8	533	585
	1:9	511	581
	1:10	519	583
TFOB	14.3%TFMB/85.7%ODA//BTDA	472	560
	14.3%TFMB/85.7%ODA//50%BPDA/50%PMDA	550	587
	14.3%TFMB/85.7%ODA//50%BTDA/50%PMDA	456	567

**Table 4 polymers-15-01256-t004:** Mechanical properties of PI films in group HFOP and group HFOB.

Monomers	Tensile Strength/MPa	Elongation at Break/%
ODA//PMDA	54.	8.289
25%HFBAPP/75%ODA//PMDA	73	18.22
50%HFBAPP/50%ODA//PMDA	72	17.32
75%HFBAPP/25%ODA//PMDA	48	11.74
HFBAPP//PMDA	50	11.05
50%HFBAPP/50%ODA//BPDA	66	10.59
50%HFBAPP/50%ODA//BTDA	65	12.82
50%HFBAPP/50%ODA//50%BPDA/50%PMDA	62	12.23

**Table 5 polymers-15-01256-t005:** Mechanical properties of PI films in group TFOP and group TFOB.

	Monomers	Tensile Strength/MPa	Elongation at Break/%
TFMB:ODA	1:4	92	23.75
	1:5	106	18.72
	1:6	110	20.23
	1:7	89	19.88
	1:8	80	21.84
	1:9	54	10.88
	1:10	59	15.29
TFOB	14.3%TFMB/85.7%ODA//BTDA	56	9.23
	14.3%TFMB/85.7%ODA//50%BPDA/50%PMDA	56	9.23

**Table 6 polymers-15-01256-t006:** Dielectric properties and hygroscopic of PI films in group HFOP and group HFOB.

Monomers	Dk (1 MHz)	Df (1 MHz)	Moisture Absorption Rate (%)
ODA//PMDA	3.03	0.0115	2.778
25%HFBAPP/75%ODA//PMDA	2.53	0.00753	1.575
50%HFBAPP/50%ODA//PMDA	2.39	0.00702	0.917
75%HFBAPP/25%ODA//PMDA	2.29	0.00645	1.034
HFBAPP//PMDA	2.21	0.00612	1.213
50%HFBAPP/50%ODA//BPDA	2.38	0.00734	1.525
50%HFBAPP/50%ODA//BTDA	2.37	0.00813	1.727
50%HFBAPP/50%ODA//50%BPDA/50%PMDA	2.40	0.00730	1.000
50%HFBAPP/50%ODA//50%BTDA/50%PMDA	2.41	0.00801	1.052

**Table 7 polymers-15-01256-t007:** Dielectric properties and hygroscopic of PI films in group TFOP and group TFOB.

	Monomers	Dk (1 MHz)	Df (1 MHz)	Moisture Absorption Rate (%)
TFMB:ODA	1:4	2.08	0.00743	1.901
	1:5	2.10	0.00715	1.569
	1:6	2.12	0.00698	1.224
	1:7	2.24	0.00756	1.504
	1:8	2.44	0.00794	1.757
	1:9	2.53	0.00815	1.825
	1:10	2.65	0.00873	2.158
TFOB	14.3%TFMB/85.7%ODA//BTDA	2.19	0.00745	1.458
	14.3%TFMB/85.7%ODA//50%BPDA/50%PMDA	2.29	0.00786	1.274
	14.3%TFMB/85.7%ODA//50%BTDA/50%PMDA	2.25	0.00762	1.128

## Data Availability

The original data of the experiment is not publicly available and can be obtained by contacting the author.

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
