# Peer review of "Reducing the Permittivity of Polyimides for Better Use in Communication Devices"

_polymers, 2023, doi:10.3390/polym15051256_

Round 1

Reviewer 1 Report

In the present manuscript the effect of three kinds of diaminesand three kinds of dianhydride on the performance of PI films were analysed. Especially in the context of 5G wireless data transfer, this investigation is of great interest. However, a significant number of problems in the present form of the manuscript have to be adressed by the authors in order to render the draft suitable for the Polymers Journal. Comments on the work can be found in the attached pdf file.

Author Response

Thank you for taking the time to provide constructive comments and comments on this article. Please refer to the attachment for details of modification.

Reviewer 2 Report

Dear Authors:

This reviewer expresses gratitude for your research on the behavior of several types of polyimide-doped mixes' thermal, mechanical, and dielectric behaviour. The impact of fluorinated groups on these attributes is this paper's key contribution. However, this analysis is limited to very low frequencies in the case of permittivity (1 MHz).

GENERAL COMMENTS:

English should be extensively reviewed since there are a high number of grammar mistakes and sometimes it is difficult to understand the concepts explained by authors. The style should be also reviewed. 

A great number of acronyms are not defined such as NVT, NPT, Tg, T1%, T5%, etcetera. The document should be self explanatory. Authors should consider that readers from the electronic engineering field could read this work and would not understand what those acronyms mean.

In section 2 authors provide both the simulation method and the simulated results. Methods and results should be separated and perhaps, these simulated results should be included in section 4.

There are numerous figures in the text that are added without any presentation or justification. Any figure must be displayed and examined properly. Please read over. Use phrases that are similar to this one: "Figure XXX shows/depicts the behavior of... "

Particularly for permittivity and hygroscopicity, I find that the experimental approaches lack sufficient description. Additionally, all measurements' uncertainty is not provided, and it ought to be shown or discussed in the figures and tables.

The authors mention high-frequency applications, such as 5G, in a number of their phrases, yet the dielectric properties shown in this paper are for extremely low frequency values. Hence they are of little use for those high-frequency technologies. Please refrain from putting such remarks in the text.

SPECIFIC COMMENTS:

The title of the paper indicates "High Frequency Communication" but the measurements of permittivity are made up to 1 MHz, which is a very low frequency in communications. The title should be changed avoiding the indication of "high frequency". Additionally, the research also investigates mechanical and thermal properties of polymers that could be mentioned in the title.

Please indicate the processor, operative system and memory used, simulation time, etcetera in the simulation section. 

Please provide definitions for NVT and NPT in section 2.1.

Figure 1 should be correctly presented. Typical wording would look like this: "Figure 1 displays data that was obtained via... The findings lead us to believe that..."

In Figure 1, please describe the differences between the various colored lines. The graph or the caption should contain legends. Figure 1 should have two different captions, (a) and (b), with their respective explanations because there are two plots.

The authors present three different polarization results, but they make no attempt to explain why or how these results are related. Is it possible to draw the same conclusions from Table 1's single result, or are the three columns required?

Please indicate stirring speed and stirrer model in section 3.2.

Figure 5 is put into the text without being presented or given any context. 

How do authors measure the loss factor? The loss factor, which is connected to the conductance of the dielectric, is not provided by equation (1); it only gives the dielectric constant. An additional equation should be provided for loss factor calculation.

Unless the authors have followed the accepted method for determining the dry mass in equation 2, mass 1 cannot be regarded as the dry mass. Typically, three weights that are the same during the drying process are regarded as the dry mass. Please specify a reference for this process. Additionally, please indicate if m1 is the dry mass.

What do Tg, and T5% mean in section 4.2? There is no definition for these terms in the text. Authors should be aware that readers from the telecommunication or electronic areas would not understand this nomenclature. These terms should be properly explained in the experimental methodology description.

In Figure 6, weight loss is plotted against temperature rather than the thermal properties of different polymer groups. Please change the captions. Figure 6 is not at all discussed in the text, nor is it explained how it was obtained. This ought to be avoided.

Please rewrite the caption for Table 3. These aren't the thermal characteristics of PI films (specific heat, thermal conductivitiy,... etcetera) but DSC parameters that should be defined)

The authors do not introduce Tables 6 and 7 before drawing conclusions. Avoid doing this by first presenting the table and outlining its contents. The conclusions are then spoken about.

The loss factor is given in several digit positions by the authors, however given the uncertainty, it is possible that these digits aren't truly representative. Please think about this and express only the representative digits. The same holds true for other tables, such as the moisture absorption rate. Uncertainty should be provided for all measurement and instruments.

I hope that these comments help authors to improve their document.

With best regards

Author Response

(The authors gave the same response as above.)

Reviewer 3 Report

According to the idea of the article, the permittivity is one of the important characteristics of the studied films. In this case, the word "permittivity" itself should probably be present in the title of the text. The title "low dielectric films..." is not entirely clear.
The authors should provide more information about the thickness of the obtained films and about the thickness spread over the sample area. The sample capacity strongly depends on the thickness of the film, therefore, accurate determination of the thickness of the film is critical for accurate calculation of the permittivity. The authors give the values of permeability with an accuracy of two decimal places. How accurately was the film thickness measured? What is the error of these measurements? What is the accuracy of the dielectric constant calculation?
In lines 208-217, errors in the placement of spaces and in the design of formula 1 should be corrected. Explanations to the formula should not be written from a new line and with a capital letter (line 209).
The text contains two fig. 5. It should be corrected.

Author Response

(The authors gave the same response as above.)

Round 2

Reviewer 1 Report

Unfortunately, a significant number of comments were either ignored or only briefly answered. The manuscript has not improved since the first version, thus I cannot recommend the publication of the work. This is unfortunate as the information provided is in general of interest. I would recommend a complete reworking of the paper and resubmission.

Author Response

Thank you for your comments and suggestions on this article, and we have replied to one by one in the attachment.

Reviewer 2 Report

To the authors:

Thank you for your efforts to make the paper better. My primary issues regarding the paper's quality have not been addressed, despite the fact that some of the proposed changes and adjustments have been made.

Usually, the results and methodology should not be mixed. They are typically described in separate parts by authors. The reviewed version of this paper does not change this. Section 2 still has a subsection describing the methodology of simulations and the results. These results should be explained in section for devoted to results.

The equation used to calculate the dielectric loss factor is not provided by the authors, nor is there any indication of the degree of uncertainty surrounding the observations they provide in the publication. This is an unacceptable method of displaying measurements. Measurement +/- Uncertainty is the typical measurement expression. The section on experimental setup should also indicate the instrument uncertainty and the appropriate references.

Authors do not describe what are the three types of polarization presented in the paper and why they present the three of them. What are the differences among them? How are these polarization indicators calThe authors do not disclose the equation used to get the dielectric loss factor, nor do they make any mention of the level of uncertainty surrounding the measurements they present in the journal. This style of measurement display is unacceptable. The standard measurement expression is measurement +/- uncertainty. The instruments uncertainty and the appropriate references should also be mentioned in the section on experimental setup .

The three types of polarization that the authors offer in the paper and their justification for doing so are not explained by the authors. What distinguishes them from one another? These polarization indicators: how are they determined?culated?

Finally, the headline claims "high frequency communication," although the frequency of the measurements is 1 MHz, which is actually a relatively low frequency for communications (usually made by cable at that frequency). Therefore, throughout the manuscript, any mention of this "high frequency" should be eliminated.

Author Response

Thank you for your comments and suggestions on this article, and we have replied to one by one in the attachment. Please refer to the latest edition of the paper for detailed modification.

Reviewer 3 Report

The text should be carefully checked for mistakes and misspells.
The quality of figures 1, 5, 6, 7, 8 is low for reading and understanding.
There are two fig. 5 in the text.
The main questions of the first review about the thickness measurements are still require a expanded answers in the text of the article.

Author Response

(The authors gave the same response as above.)

Round 3

Reviewer 1 Report

Now, most of the comments from my FIRST review have been addressed. Next time, a rebuttal letter with responses to the reviewers comments would help, instead of simply resubmitting an updated version of the paper.

Author Response

Thank you for your comments and suggestions on this article. Please refer to the attachment for details on the modification.

Have a goodweekend。

Reviewer 2 Report

Almost all of my questions and suggestions were met with satisfactory responses from the authors.

However, the parallel plate technique equation for measuring dielectric loss has not been provided. My recommendation is that they add a new equation (2) and that they explain that they use this parallel plate technique:

Df= d /(w*Rp*e0*S)  (2)

where w is the angular frequency (rad/s) and Rp is the parallel plate equivalent parallel resistance.

The English style, on the other hand, still needs a lot of work

Author Response

Thank you for your comments and suggestions on this article. Please refer to the attachment for details on the modification.

Have a good weekend.

Reviewer 3 Report

The authors have improved the text during two revisions and have answered questions and negative comments. Article can be published in present form.

Author Response

Thank you for your comments and suggestions on this article.

 Have a great weekend